# Association between Hip/Groin Pain and Hip ROM and Strength in Elite Female Soccer Players

**DOI:** 10.3390/jcm13185648

**Published:** 2024-09-23

**Authors:** Eloy Jaenada-Carrilero, Luis Baraja-Vegas, Paula Blanco-Giménez, Raul Gallego-Estevez, Iker J. Bautista, Juan Vicente-Mampel

**Affiliations:** 1Doctoral School, Catholic University of Valencia Saint Vincent Martyr, 46900 Valencia, Spain; eloy.jaenada@ucv.es; 2Faculty of Science Health, Physiotherapy Department, Catholic University of Valencia Saint Vincent Martyr, C/Ramiro de Maetzu 14, 46901 Torrent, Spain; paula.blanco@ucv.es (P.B.-G.); i.bautista@chi.ac.uk (I.J.B.); juan.vicente@ucv.es (J.V.-M.); 3Faculty of Science Health, Podiatry Department, Catholic University of Valencia Saint Vincent Martyr, C/Ramiro de Maetzu 14, 46901 Torrent, Spain; raul.gallego@ucv.es; 4Institute of Sport, Nursing and Allied Health, University of Chichester, Chichester PO19 6PE, UK

**Keywords:** women’s football, groin pain, hip injuries, muscle strength, articular range of motion

## Abstract

**Background/Objectives**: Hip strength and range of motion have been compared in soccer players with and without hip and groin pain but only in male footballers or gender-combined samples. In female soccer players, the biomechanics contributing to this injury remain poorly understood compared to other sporting injuries. The aim of the present study is to investigate whether differences exist in adductor and abductor isometric test values and hip joint range of motion between elite female soccer players with longstanding groin pain and injury-free controls. **Methods**: Ten female elite soccer players with current longstanding hip and groin pain and twenty-five injury-free controls from the same teams were included in the study. Hip adductor and abductor isometric strength were evaluated with a hand-held dynamometer. A bent knee fall-out test was also utilized to examine the hip joint range of motion. **Results**: A significant difference in abductor isometric test values was observed between the control group (2.29 ± 0.53 N/Kg) and the hip and groin pain group (2.77 ± 0.48 N/Kg; *p* = 0.018). Furthermore, the injured group showed a decreased adductor/abductor ratio compared to the control group (1.00 ± 0.33 vs. 1.27 ± 0.26; *p* = 0.013). No differences were observed in the bent knee fall-out test (*p* = 0.285). **Conclusions**: Female elite soccer players with current longstanding hip and groin pain exhibited higher abductor isometric strength and lower adductor/abductor ratio compared to non-injured women players. There were no differences in the BKFO test between groups.

## 1. Introduction

Hip and groin pain (HGP) is common in field sports characterized by repetitive kicking, accelerations, decelerations, jumps, and changes of direction [1,2], and symptoms are often longstanding [3]. During a competitive season, groin problems are reported by over 59% of male soccer players and 45% of female soccer players [4]. In elite male soccer players, hip and groin injuries account for 14–20% of time-loss injuries and over 10% in sub-elite male soccer players [5]. Elite female players had a 3-times lower risk of reporting groin problems as compared with elite male players [4]. Despite this, Thorarinsdottir et al. [6], in a recent study, found that the incidence rate and burden of groin injuries in elite female soccer players were substantial, with 1.6 injuries/1000 h and 11 days lost/1000 h, respectively. Langhout et al. [7] revealed in 2019 that hip and groin pain was the most common non-time-loss injury in female amateur soccer players, with sub-elite female players exhibiting an overall prevalence of hip/groin injury of 27%.

Recent systematic reviews compiled the findings of studies investigating the risk factors for HGP [8,9,10]. These reviews concluded that previous injuries, male sex, and diminished adduction strength (absolute and relative to abduction strength) were significant risk factors and described conflicting results for hip range of motion. A systematic review in 2017 [11] investigated exclusively the association between hip range of motion and groin pain and concluded that reduced total hip rotation was consistently related to groin pain.

Reduced hip adductor strength has been identified as the principal intrinsic factor associated with hip and groin pain in male adult soccer players [12,13,14]. In pre-season testing, hip adductor strength was up to 5.4% weaker in players with previous-season groin pain than in those without [13], while higher levels of pre-season hip adductor strength have been identified as a protective factor against HGP [12]. Moreover, significantly reduced hip adductor-to-abductor muscle strength ratios (less than 0.8/1) have been reported in adult football players with groin problems [15,16]. In fact, adductor strengthening is the main intervention to prevent and treat hip/groin injuries in adult male soccer players [17]. 

The relationship between hip and groin pain and hip range of motion (HROM) remains complex. Some studies have identified a correlation between restricted HROM and the presence of HGP in athletes [11,18,19]. In contrast, others have yielded inconclusive results, failing to establish any relationship [20,21]. Currently, there is increasing interest in the role of femoroacetabular impingement (FAI) because it has been demonstrated to be associated with reduced internal rotation (IR) of the hip joint [22] and HGP [23]. 

The findings in the literature comparing hip strength and range of motion in soccer players with and without HGP were made in male footballers or in gender-combined samples. Furthermore, hip biomechanics have not been documented in female soccer players with longstanding hip and groin pain. Therefore, in female soccer players, the biomechanics contributing to HGP remain poorly understood and under-investigated compared to other sporting injuries, such as anterior cruciate ligament injury [24,25] or patellofemoral pain syndrome [25,26]. A better understanding of adductor and abductor isometric values and hip joint ROM in female soccer players with longstanding groin pain compared to injury-free controls is needed. The aim of the present study is to investigate whether differences exist in adductor and abductor isometric test values and hip joint range of motion between elite female soccer players with longstanding groin pain and injury-free controls. We hypothesize that the HGP group will show lower isometric adduction strength, lower isometric adductor/abductor ratio, and lower hip ROM than the control group based on the study by Nevin et al. [19].

## 2. Materials and Methods

### 2.1. Study Design

An observational study with a case-control design was conducted to evaluate differences in hip adductor and abductor strength and hip ROM in elite female soccer players with and without hip and groin pain. No published research has investigated hip biomechanical differences in female soccer players with and without hip and groin pain. This study was approved by the Ethics Committee of the Catholic University of Valencia (reference number: UCV/2021-2022/150). All participants who chose to enroll in the study provided their agreement by signing an informed consent form prior to participation in accordance with the ethical guidelines of the Declaration of Helsinki [27]. The trial adhered to the STROBE guidelines for both the design and progression of the participants [28].

### 2.2. Participants 

A sampling strategy for participant recruitment was defined for the case-control study. All the individuals constituted the population of interest. The case was considered elite female soccer players with hip and/or groin pain, recruited by means of referral from clinicians, while control was considered healthy elite female soccer players. The participants belonged to several elite soccer teams in Spain. Players were categorized as elite based on the participant classification system elaborated by McKay et al. [29] (i.e., national first division and international championships; 10 h of training per week). The participants were recruited between January 2021 and December 2023 in the in-season period. The inclusion criteria for injured soccer players were as follows: (i) current hip or groin pain as described in the Doha Agreement [30], which has demonstrated excellent inter-examiner reliability [31]; (ii) due to sport (iii); at least six weeks duration. The exclusion criteria were as follows: (i) palpable inguinal or femoral hernia; (ii) clinical signs or symptoms of urinary tract infection; (iii) lumbar radiculopathies [32]; (iv) clinical suspicion of nerve entrapment syndrome; (v) gynecological disorders. The criteria were assessed via direct interviews and reliable, standardized clinical examination [31]. Injury-free controls were recruited from the same teams in the same period.

### 2.3. Sample Size

In the absence of previously published studies in women for sample size calculation, we based our calculations on the reference variance from the study by Malliaras et al. [33], which compared two related means. To achieve a power of 0.80 and a bilateral with 2 groups α level of 0.05, we aimed to detect a variance of 2. Considering a 10% loss to follow-up, we calculated a total sample size of 28 participants with an allocation ratio of N2/N1 = 3/1 (7 in the case group and 21 in the control group). The sample size was determined using GPower^®^ software (Franz Faul, Universität Kiel, Kiel, Germany), version 3.1.9.2.

### 2.4. Primary Outcomes

#### 2.4.1. Measurement of Anthropometric Variables Measurements

On the first day of the study, participants were interviewed to gather anthropometric data. Both weight and height were measured using a scale with an integrated stadiometer.

#### 2.4.2. Range of Motion 

Hip joint range of motion was assessed using the bent knee fall-out test (BKFO), which is used to measure hip ROM combining hip flexion, abduction, and external rotation [20,34]. For the BKFO test, the athlete was positioned in a supine position with their hips flexed to 45 degrees and knees flexed at 90 degrees, as confirmed by a universal goniometer. The athlete’s feet were kept together. The athlete was instructed to allow both knees to fall outward while keeping their feet together. The investigator applied gentle overpressure to verify that the player had fully relaxed at the limit of movement. The distance between the most distal point on the head of the fibula (marked with indelible ink) and the surface of the table was measured by the same therapist using an inflexible tape measure.

Previous studies reported excellent reliability of the BKFO distance with an interrater ICC of 0.89 and an interrater ICC of 0.91. The SEM is 1.0 when using this protocol [19].

#### 2.4.3. Strength 

All hip strength assessments were conducted by the same physiotherapist with an experience of 15 years to mitigate potential errors. A portable hand-held dynamometer (HHD) (MicroFET 2, Hoggan Scientific, LLC, Salt Lake City, UT, USA) was used for hip strength assessments, with calibration performed before testing. The calibration procedure involved setting the device to zero. Additionally, at the start of each season, the HHD was recalibrated using a known load to maintain consistent and accurate measurements over time. Maximal voluntary isometric hip adduction and hip abduction force in both dominant and non-dominant legs were tested. All assessments were performed using a massage table. The order in which the tests were conducted varied systematically among participants. To eliminate or reduce potential biases arising from the order of tests and any transference effects that might occur if one test influenced subsequent performance, we implemented a specific sequence. Each player completed a series of tests in the order of A, B, B, A, where ‘A’ represents the abductor test and ‘B’ denotes the adductor test. Following this, the average of the two ‘A’ conditions was calculated, and the same procedure was applied to the ‘B’ conditions. Two sub-maximal familiarization trials were carried out to ensure the players correctly performed the action of pushing into the belt and HHD. Verbal encouragement was provided during the test with the standard instruction, “push, push, push”.

Isometric hip adduction and abduction were examined in standardized positions based on previously used reliable methods [35,36]. The participants were positioned supine and instructed to stabilize themselves by holding the sides of the table with their hands. For adductor measurement, the examiner (E.J.) applied resistance in a fixed position, 2 cm proximal to the edge of the medial malleolus. The abductor measurement was conducted using the HHD, with a belt fixation positioned proximal to the edge of the lateral malleolus [37]. The participant exerted a maximum isometric voluntary contraction against the HHD for 5 s. Two trials were conducted on the same leg, with a 30-s rest period between repetitions. The mean peak force (measured in Newtons [N]) recorded for each limb across two attempts was used for data analysis. If the variability between trials was greater than 10%, a new trial was performed. Absolute values (N) from each player were normalized to body mass (N/kg). Preliminary results showed excellent reliability coefficients (ICC = 0.92 to 0.96) and nearly perfect validity scores (r = 0.996) of this procedure in comparison to the fixed-frame dynamometry system [38].

### 2.5. Statistical Analysis

All results were expressed as mean and standard deviation (±SD). Normality and homogeneity of variance assumptions were analyzed using the Shapiro–Wilk test and Levene test, respectively. Relative reliability was examined using ICC2.k, whereas absolute reliability was calculated using SEM [39]. The assumption of normality was assessed using the Kolmogorov–Smirnov test. In addition, the assumption of homogeneity of variance was calculated using Levene’s test. The significance level was set at *p* < 0.05. Analysis of variance (ANOVA) was used to analyze the differences among the group means in a sample. To analyze the differences between groups, a main effect interaction group*dominance was performed with the groups (i.e., “hip and groin pain” and “healthy”). Post hoc tests utilizing Tukey’s correction were conducted to address multiple comparisons. All post hoc analyses were presented using mean differences (MD) and 95% confidence intervals (CI 95%). Effect size (ES) was calculated according to Cohen formulas [40] and considered trivial (<0.20), small (0.20–0.59), moderate (0.60–1.19), large (1.20–1.99), and very large (>2.00) [41]. The statistical significance level was set at *p* < 0.05. All calculations were performed using a statistical analysis tool (JASP v.0.17.1, The Netherlands). Statistical analysis was conducted by a researcher who was not involved in any of the phases of data collection and received data in coded form.

## 3. Results

### 3.1. Participation Flow and Sample Characteristics

A total of 42 subjects were enrolled in the study (12 injured and 30 healthy controls). The reasons for exclusion at each stage are shown in the flowchart (Figure 1).

Ten female elite soccer players with current hip or groin pain (age = 23.2 ± 3.85 years, body mass = 58.4 ± 6.05 kg, height = 1.65 ± 0.05 m) volunteered to participate in the study. Twenty-five injury-free controls (age = 24.5 ± 5.08 years, body mass = 61.0 ± 6.05 kg, height = 1.68 ± 0.04 m) were recruited from the same soccer clubs. The baseline anthropometric variables are shown in Table 1.

### 3.2. Main Outcomes

#### 3.2.1. Descriptive Analysis

The primary outcomes are shown in Table 2. The total number of participants in the study and detailed descriptive values by group are provided. Strength measures could not be recorded for 2 control players, and BKFO was not recorded for an additional 3 control players, resulting in 10 participants in the symptomatic group and 25 in the control group.

#### 3.2.2. Comparations Main Outcomes per Group Isometric Hip Strength, Adduction/Abduction Ratio, and BKFO

##### Isometric Hip Adduction Test

Regarding isometric hip adduction, repeated measures ANOVA showed no statistically significant differences between groups (F[1, 33] = 0.86, *p* = 0.359, η^2^*p* = 0.026). Furthermore, in the main effect of Group*Dominance interaction, no statistically significant differences were found (F[1, 33] = 0.228, *p* = 0.636, η^2^*p* = 0.007) (See Figure 2A).

##### Isometric Hip Abduction Test

Regarding isometric hip abduction, repeated measures ANOVA showed statistically significant differences between groups. (F[1, 33] = 6.17, *p* = 0.018, η^2^*p* = 0.158) in favor of the HGP group. Regarding the main effect of Group*Dominance interaction, tendential but no statistically significant differences were found (F[1, 33] = 3.72, *p* = 0.062 η^2^*p* = 0.101). Thus, Tukey’s post hoc analysis for leg dominance factor (mean [SE, *p*, t, ES]) found that injured soccer players displayed stronger values on the non-dominant side in comparison with the non-dominant side (0.644 [0.211, 0.019, 3.054, 1.143]) and dominant side (0.571 [0.211, 0.045, 2.710, 1.014]) in control players (See Figure 2B).

##### Isometric Hip Adduction/Abduction Ratios

Statistically significant differences were observed in the isometric hip adduction/abduction ratio in the repeated measures ANOVA (F[1, 33] = 6.909, *p* = 0.013, η^2^*p* = 0.173) in favor of the control group. In the effect of Group*Dominance interaction, no statistically significant differences were found (F[1, 33] = 2.744, *p* = 0.107, η^2^*p* = 0.077). Tukey’s post hoc analysis for leg dominant factor (mean [SE, *p*, t, ES]) showed statistically significant differences in the adduction/abduction ratio in the non-dominant side in injured players compared with the non-dominant side in control players (−0.370 [0.119, 0.016, −3.102, −1.161]) in favor of control players (See Figure 2C).

##### Bent Knee Fall-Out Test

With respect to BKFO, the results of repeated measures ANOVA showed no statistically significant differences between the injured and healthy groups (F[1, 33] = 1.181, *p* = 0.285, η^2^*p* = 0.035). Moreover, no significant differences were found in the effect of Group*Dominance interaction (F[1, 33] = 0.055, *p* = 0.817, η^2^*p* = 0.002).

## 4. Discussion

The present study aims to evaluate possible differences in the isometric strength values of hip adductors, hip abductors, and the ratio adductor/abductor, and hip mobility assessed by the BKFO test between elite female soccer players with and without hip and groin pain. This study is the first to investigate the biomechanical differences in elite female soccer players with and without hip and groin pain. Previous studies have analyzed female soccer players [35,36,42], but never at the elite level and in isolation. In our study, the group of injured players was found to have greater abductor strength than the group of healthy players. Players with hip and groin pain had significantly greater abductor strength on the non-dominant side compared to the dominant and non-dominant side in control players. Injured players had reduced adductor strength values compared to the control group, but the differences were not significant. On the other hand, significant differences were observed in the adductor/abductor ratios between groups, with the healthy players showing significantly higher ratios. Regarding hip rotation ROM, the BKFO test revealed no significant differences between injured and control players.

The evaluation of adductor isometric strength in soccer players experiencing HGP is the most commonly employed test in the literature to examine differences between players with and without pain [13,19,35,36,42,43,44,45]. Previous findings have shown that players with HGP exhibited diminished isometric strength values in their hip adductors. The present investigation does not support those findings as, although the injured players did show reduced isometric adductor strength, the values were not significant. This may be explained either by the low sample size or by the difference in the population sampled, being exclusively female, or perhaps also by the fact that the players with groin pain also included players with hip pain.

The findings of increased isometric strength in the abductors of the injured players are in line with the results obtained by Schoffl et al. [36], who observed that increased hip abductor strength was associated with a previous history of hip and groin pain. This study has the highest percentage of women in its sample, with 47 women out of 105 total subjects, of whom 6 had a history of HGP, and 5 developed it during the post-testing season. These results in women could be explained by the presence of compensatory movement strategies during the injury or because the hip abductor musculature is more prone to strength fluctuations in female athletes. The relationship between abductor weakness and the risk of LCA injuries [46] or patello-femoral syndrome injury [47,48] has been extensively studied.

Our results of a significantly reduced adductor/abductor ratio in the HGP group coincide with previous studies in male samples [36,49]. However, it is important to highlight that the ratio of 1.27 obtained in the control female soccer players is almost three-tenths higher than that obtained in professional healthy male soccer players in previous studies [34]. A ratio of less than 0.8/1 has been determined as a risk factor for HGP [16]. The ratio obtained in our sample of pain-free women players could explain the lower incidence of HGP in female soccer players compared to male players, but the small sample size limits these results.

The similar BKFO values between both groups obtained in this study support the results obtained by Mosler et al. [20] and Tak et al. [18], in which no differences were found between injured and controls. In contrast, Nevin et al. [19] did obtain significantly higher BKFO values, i.e., lower ROM, in players with hip and groin pain. It should be noted that the sample in this study was exclusively male and involved in Gaelic soccer (a form of football similar to rugby). Today, there is still controversy as to whether ROM limitation is a risk factor related to HGP. The recent review of Tak et al. [11] concluded that reduced total hip rotation ROM is the factor most strongly related to groin pain in athletes.

The present study is not without limitations. The sample of control subjects was 25 players, while the number of injured subjects was only 10 players, which required data from three professional teams in four different seasons. This is explained by the low incidence of HGP in women. Studies involving more elite female soccer players are recommended to obtain more representative data. The investigator was not blinded with respect to group assignment. Although a strict protocol was followed in all tests to minimize the risk of differential behavior between the examiner and participants, the lack of blinding may have biased our results.

Further research employing a rigorous blinding protocol is needed to substantiate our findings. This research evaluated only ROM using the validated BKFO test in which hip flexion, abduction, and external rotation are analyzed. Future research would be necessary to evaluate whether differences exist between transverse, sagittal, and coronal plane movements of injured and uninjured women footballers. It would also be important to record other extrinsic factors, such as pitch position, or intrinsic factors, such as lower limb length. 

The cross-sectional design of the present study poses a limitation. Future prospective studies are necessary to gain a deeper understanding of the temporal relationship in the development of the observed deficits in abductor strength and adductor/abductor ratio. Longitudinal research is also needed to explore how adductor and abductor strength test values and hip joint ROM change upon return to sport. In addition, further research evaluating female soccer players with and without groin pain in performing a battery of motor control tests is warranted to explore how ROM and hip strength influence functional task performance in women.

## 5. Conclusions

Female elite soccer players with current longstanding hip and groin pain exhibited higher abductor isometric strength and lower adductor/abductor ratio compared to non-injured women players. There were no differences in the BKFO test between female elite soccer players with and without hip and groin pain. Prospective research and larger samples will be needed to confirm these results.

## Figures and Tables

**Figure 1 jcm-13-05648-f001:**
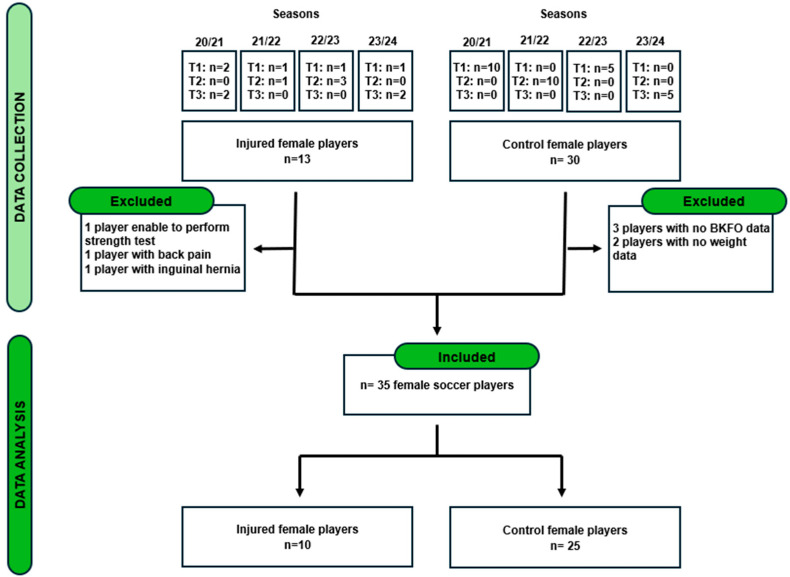
Flow chart for inclusion into the study.

**Figure 2 jcm-13-05648-f002:**
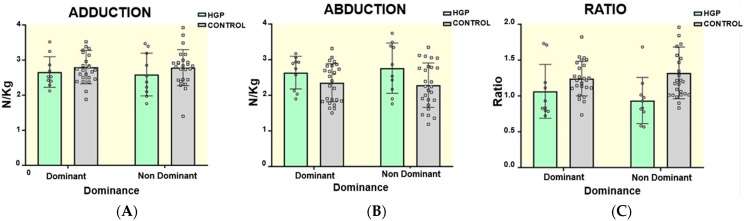
**Differences in strength between groups** (**A**) Isometric hip adduction and (**B**) abduction (**C**) and adduction/abduction ratio results for the dominant and non-dominant leg for HGP and healthy female soccer players. The whiskers represent a 95% confidence interval.

**Table 1 jcm-13-05648-t001:** Descriptive statistics (mean and standard deviation) of the anthropometric variables analyzed were separated by case-control group.

Variables	HGP Players	Control Players	*p*-Value
Participants (*n*)	10	25	nc
Age (years)	23.2 (3.85)	24.5 (5.08)	0.48
Height (m)	1.65 (0.05)	1.68 (0.04)	0.06
Body mass (kg)	58.4 (6.05)	61.0 (5.84)	0.25
Dominance (R/L)	8/2	24/1	nc

R = right; L = left; nc = not calculated.

**Table 2 jcm-13-05648-t002:** Descriptive statistics of isometric hip strength, adduction/abduction ratio and BKFO.

Variables	HGP Players	Control Players
*Add d* (N/kg)	2.66 ± 0.44	2.80 ± 0.48
*Add nd* (N/kg)	2.59 ± 0.61	2.78 ± 0.51
*Med add* (N/kg)	2.62 ± 0.48	2.79 ± 0.48
*Abd d* (N/kg)	2.64 ± 0.46	2.32 ± 0.53
*Abd nd* (N/kg)	2.89 ± 0.62	2.25 ± 0.61
*Med abd* (N/kg)	2.77 ± 0.48	2.29 ± 0.53
*Ratio d*	1.06 ± 0.38	1.24 ± 0.24
*Ratio nd*	0.94 ± 0.32	1.30 ± 0.36
*Med ratio*	1.00 ± 0.33	1.27 ± 0.26
BKFO d (cm)	20.2 ± 2.58	18.5 ± 4.63
BKFO nd (cm)	21.0 ± 2.44	19.6 ± 5.08

d = dominant, nd = non dominante, Med = media.

## Data Availability

The data presented in this study are available on request from the corresponding author because this study is part of a doctoral thesis.

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
