# Peer review of "Association between Hip/Groin Pain and Hip ROM and Strength in Elite Female Soccer Players"

_jcm, 2024, doi:10.3390/jcm13185648_

Round 1
Reviewer 1 Report
Comments and Suggestions for Authors
Overall evaluation: The authors have addressed a very interesting topic. There are some elements that should be clarified and expanded in order to improve the article. Some limitations, which the authors have promptly pointed out, affect the final conclusions, especially the fact that the athletes were not followed prospectively and the extremely small sample size.
Abstract: It might be useful to include the specific values of the adductor ratio for both the control group and the HGP group.
Introduction: There is a lack of an overview regarding the evaluation, treatment, and prevention of patients with HGP. The authors' hypothesis is not specified; I suggest you implement this section in accordance with the STROBE guidelines.
Materials and Methods: No information is provided about the setting and characteristics of the athletes involved: How often do they train? How often do they compete? At what level? What are the loads they are subjected to during the season? At what point in the season was the evaluation carried out? Obviously, this data can impact the study, as evaluating pre-season, in-season, or off-season yields different results. Please report how potential confounding factors, biases, and missing data were managed.
Results: Indeed, the adductor to abductor ratio values are significant between the two groups, but the ratio is still greater than or close to 1, while in studies on male populations, the values are well below 1. This should be discussed further.
Discussion: To give the study more prominence, the potential clinical impact of these results should be emphasized: What is the current practice in evaluating patients with HGP? Is there any prevention? How are they treated? This aspect could be addressed in the introduction, and then in the discussion, the study's contribution to current practice could be highlighted so that the reader better understands the motivations behind conducting this study and its clinical impact. It emerges that some tests, such as BKFO, while isometric tests, could be more useful. The association expressed from line 307 to line 314 may be forced: the authors did not evaluate the eccentric strength domain. The limitations of the study, which you have appropriately reported, reduce the possibility of generalizing the results. Please clearly specify this in the discussion.
There are several typo errors, such as spaces after citations, etc. I ask you to review the English and check for any typos.
I ask you to review the good work conducted in order to improve it overall.
Best regards
Comments on the Quality of English LanguageThere are several typos in the text, some double periods, and no space after the citations. It should be reviewed
Reviewer 2 Report
Comments and Suggestions for Authors
Dear Authors, please consider my comments below:
1. Others:
-were FAI excluded?
-was there any radiological assessment prior to the study?
-Authors mentioned low back pain as an exclusion criteria, but it must be added both to the Introduction section (consider citation: https://pubmed.ncbi.nlm.nih.gov/34200510/ ) and limitations as well;
-was there any clinical examination before enrollment, including (1) limb length, (2) hamstrings/quad strength, (3) load distribution, (4) femoral/genitofemoral/saphenous/obturator nerves assessment?
-was position on the pitch taken into account?
2. The sample size of 10 injured players limits the generalizability of the findings. Was a power analysis performed to confirm that the study is adequately powered to detect significant differences?
3. The cross-sectional nature of the study does not allow for conclusions about causality between strength/ROM deficits and injury. Will the authors conduct a prospective study to follow athletes over time to better assess causality?
4. Contrary to much of the literature, the study found no significant differences in hip adduction strength between injured and control groups. Could the lack of significant findings be due to the inclusion of both hip and groin pain cases, and would separating these subgroups clarify the results?
5. The study observed increased abductor strength in injured players, which contrasts with established findings. Could compensatory mechanisms during injury explain the increased abductor strength in the injured group?
6. The study did not blind the examiners during strength testing, which may introduce bias. Could future studies benefit from blinding to reduce potential biases in strength assessments?
7. The BKFO test may not be sensitive enough to capture differences in hip range of motion between groups. Would the inclusion of other ROM tests, such as isolated internal/external rotation, provide more robust data?
8. The study does not assess how strength and ROM deficits affect soccer-specific performance, which limits the practical application of the findings. Would incorporating functional movement tests help to better understand the real-world implications of the observed biomechanical differences?
Regards,
Comments on the Quality of English Language
minor corrections needed
Round 2
Reviewer 1 Report
Comments and Suggestions for Authors
Thank you for incorporating the requested changes. Excellent work.
Reviewer 2 Report
Comments and Suggestions for Authors
Thank you for replying. Please enhance the limitations section where applicable and focus on the language site.
Comments on the Quality of English Languageminor corrections